# The Effect of Bimanual Intensive Functional Training on Somatosensory Hand Function in Children with Unilateral Spastic Cerebral Palsy: An Observational Study

**DOI:** 10.3390/jcm12041595

**Published:** 2023-02-17

**Authors:** Catherine V. M. Steinbusch, Anke Defesche, Bertie van der Leij, Eugene A. A. Rameckers, Annemarie C. S. Knijnenburg, Jeroen R. J. Vermeulen, Yvonne J. M. Janssen-Potten

**Affiliations:** 1Adelante Rehabilitation Centre, 6301 KA Valkenburg, The Netherlands; 2Research School CAPHRI, Department of Rehabilitation Medicine, Maastricht University, 6229 ER Maastricht, The Netherlands; 3Adelante Centre of Expertise in Rehabilitation and Audiology, 6432 CC Hoensbroek, The Netherlands; 4Paediatric Rehabilitation, Biomed, Faculty of Medicine & Health Science, Hasselt University, 3500 Hasselt, Belgium; 5Department of Neurology, Maastricht University Medical Centre+, 6229 ER Maastricht, The Netherlands; 6Research School Mental Health and NeuroScience, Maastricht University Medical Centre+, 6229 ER Maastricht, The Netherlands

**Keywords:** cerebral palsy, upper limb, sensory function, bimanual intensive functional therapy, hand function

## Abstract

(1) Background: Next to motor impairments, children with unilateral spastic cerebral palsy (CP) often experience sensory impairments. Intensive bimanual training is well known for improving motor abilities, though its effect on sensory impairments is less known. (2) Objective: To investigate whether bimanual intensive functional therapy without using enriched sensory materials improves somatosensory hand function. (3) Methods: A total of twenty-four participants with CP (12–17 years of age) received 80–90 h of intensive functional training aimed at improving bimanual performance in daily life. Somatosensory hand function was measured before training, directly after training, and at six months follow-up. Outcome measures were: proprioception, measured by thumb and wrist position tasks and thumb localization tasks; vibration sensation; tactile perception; and stereognosis. (4) Results: Next to improving on their individual treatment goals, after training, participants also showed significant improvements in the perception of thumb and wrist position, vibration sensation, tactile perception, and stereognosis of the more affected hand. Improvements were retained at six months follow-up. Conversely, proprioception measured by the thumb localization tasks did not improve after training. (5) Conclusions: Intensive functional bimanual training without environmental tactile enrichment may improve the somatosensory function of the more affected hand in children with unilateral spastic CP.

## 1. Introduction

Children with cerebral palsy (CP) often have motor and sensory impairments which negatively impact upper limb function, causing limitations in daily activity and participation [1,2,3,4]. Intensive goal-oriented upper limb therapies are effective in promoting bimanual performance and daily functioning [5,6]. These rehabilitation approaches mainly focused on motor deficits. These days, however, there is increased attention on sensory impairments because of the clear link between somatosensory impairment and poor hand function in children with CP [7]. Conversely, the effects of rehabilitation interventions on changes in somatosensory hand function in children with CP are relatively understudied [8]. The studies that have been published show varying effects. Charles et al. [9] reported an improvement in two-point discrimination after constrain-induced movement therapy (CIMT) in three children with unilateral spastic CP. The less affected upper extremities of these children were constrained six hours daily for 14 consecutive days. Charles and colleagues attributed this improvement in tactile discrimination to an increase in tactile input and its subsequent change in cortical receptor fields for the fingers. Matusz et al. [10] studied the effect of hybrid CIMT on somatosensory hand function in ten children with CP. Their intervention encompassed 120 h of CIMT, 22 h of goal-directed training and shaping, and 10 h of fine motor activities that involved sensory feedback components such as temperature, texture, light and deep pressure, and vibration. Reliable differences in stereognosis, grip, and pinch tests were revealed between the more affected and less affected hands before but also after CIMT. However, only grip strength in the more affected hand was influenced by CIMT. Another study on the effects of hybrid CIMT is the study of Jobst et al. [11]. To optimize potential changes in somatosensory function, somatosensory activities were enhanced within the CIMT protocol, focusing on tactile, stereognosis, and proprioceptive modalities. They detected a significant change in tactile registration in the affected hand, but not in other aspects of clinical sensory functioning, i.e., 2-point discrimination, proprioception, and kinesthesia. Maitre and colleagues [12] reported on the effects of a multi-component intervention in infants (*n* = 37) aged 6 to 24 months. They studied somatosensory processing using cortical event-related potential (ERP) responses for tactile stimulation of the more affected hand at the contralateral and ipsilateral frontal scalp regions. Their intervention improved somatosensory processing. Kuo et al. [13] investigated the effect of a 90 h standardized hand-arm bimanual training program (HABIT), with and without tactile training, in twenty children with unilateral spastic CP. They concluded that tactile spatial resolution can improve after bimanual training and that intensive bimanual training alone, or with the incorporation of materials with a diversity of shapes/textures, may drive these changes. An intensive motor skill learning intervention involving both the upper- and lower-extremities (HABIT-ILE), without a sensory enriched environment, showed an improvement in stereognosis in the more affected hand in CP subjects, but no significant change in tactile spatial discrimination [14].

In the present study, the effectiveness of a 15-day functional, intensive, goal-oriented, clinical therapy program, focused on improving bimanual performance in daily activities in children and adolescents with CP, aged between 11–20 years, on somatosensory performance is investigated. We hypothesize that somatosensory function may improve after intensive training, even without the addition of sensory-enriched material.

## 2. Materials and Methods

### 2.1. Patients

A convenience sample of children and adolescents, diagnosed with CP, who participated in a bimanual intensive functional training (BIMT) program for the first time between 2017 and 2021, was used. Inclusion criteria were: Gross Motor Function Classification System (GMFCS) I-IV (able to stand/transfer independently), Manual Ability Classification System (MACS) I-III (able to perform, at least partially independent, on manual tasks), between 11–20 years of age, unilateral or asymmetric bilateral CP (spastic/dyskinetic/ataxic), and having clear treatment goals regarding bimanual performance tasks. Exclusion criteria were: unable to sleep over at Adelante Paediatric Rehabilitation Centre for 15 days and severe cognitive impairments that hinder active participation in the program. Inclusion for BIMT was performed by a rehabilitation physician and his/her rehabilitation team.

We tried to retrospectively collect the brain MRI of each participant because children with periventricular leukomalacia (PVL) lesions have significantly better hand function and sensation scores than children with cortical-subcortical/middle cerebral artery (MCA) lesions [15]. For the classification of the type of lesion, we used the classification introduced by Himmelmann et al. [16]. Informed consent from parents and participants was obtained.

### 2.2. Bimanual Intensive Functional Training

The BIMT program is a 15-day clinical intervention for children and adolescents, developed by researchers and clinical staff of Adelante Paediatric Rehabilitation Centre in Houthem, the Netherlands. All the staff have over seven years of experience working with children with unilateral spastic CP and are specialized in hand function problems in these children. The program is based on motor learning principles and functional therapy, according to the (inter)national guidelines in CP, i.e., the therapy is goal-directed, using a context-based approach, aimed at the active participation of the child, focusing on activity and participation, and incorporating parent involvement. The program focuses on using both hands in numerous everyday two-handed skills. The affected hand is considered the assisting hand in a stabilizing or supporting role. Potential candidates are extensively screened before participation by members of our expert team. During this screening, every participant formulated their treatment goals and needs. Together with the therapist and parents, these goals were ranked in the top three. The performance of these three goals was assessed, and after a task analysis, they were translated into individualized goal-directed therapy sessions by experienced therapists. All training was performed on-site. To encourage training intensity, all participants were paired with a personal buddy who continuously prompted the participant to use both hands when performing activities throughout the day. These buddies were interns in occupational therapy, movement sciences, medicine, or sports training who followed a one-day training program by experienced therapists. During the program, the students were supervised by experienced therapists. In terms of manpower, one healthcare professional supervised three participants.

Therapy starts from the moment of waking up, when self-care activities such as dressing and grooming are practiced. As for breakfast activities, participants have to set the table, prepare their breakfast, and clean up afterwards. All breakfast items are chosen in such a way that two hands are needed to spread the bread, pour the milk, or get the yogurt out of the package, for example. The breakfast session is followed by a therapy block of 90–120 min in which the specific personal treatment goals are trained. In the afternoon, activities that specifically focus on personal goals are alternated with group activities (survival, sports, bimanual gaming, and recreational activities). The participants are also involved in preparing lunch and dinner. To improve retention of the trained skills in the home environment after the therapy program, in the intervening weekend, participants train on personal goals together with their caregivers. This way, participants receive a daily total of 6–7 h of intensive therapy, totaling 80–90 h for the entire program.

Before participating in this clinical BIMT program, these children received therapy as usual. According to the Dutch guidelines for the treatment of children with CP, this amounts to 30 min of therapy once or twice a week. Treatment is given based on requests for help from the parents and/or the children themselves, or is based on the rehabilitation team’s findings during regular examinations.

### 2.3. Somatosensory Function Testing

The tactile perception was measured using a monofilament task (MFT): We used the 6.65 Semmes-Weinstein monofilament (SWM) [17] and tested nine palmar areas of the affected hand in random order, with vision blocked. Participants were asked if they felt the monofilament by saying ‘yes’, and, subsequently, to point out the location of touch with their less affected hand while vision was restored. We chose to only use the 6.65 monofilament to reduce testing time and the burden to the participant. With this, we deviated from the original protocol as described by Bell-Krotoski [18]. When participants were able to identify the monofilament touch within a range of two centimeters, the score was 1; when identified over two centimeters from the tested site, the score was 2. Participants scored 3 if no administered stimulus was identified. A few practice trials were given on the less affected hand until the procedure was understood. The MFT can be reliably performed in the vast majority of children aged four years and above [19] and is recommended for assessing tactile function in children with cerebral palsy [20].

Vibration sense was measured using a vibration task (VT): A 128 Hz tuning fork was placed on 18 areas of the affected hand. Participants had to report whether a vibration was recognized in these 18 areas. The number of reported areas resulted in a final score (i.e., score 0–18) [21].

Stereognosis was measured using a stereognosis task (SGT): Participants were asked to identify three familiar items (marble, button, key) without vision after the assessor had placed the item in the affected hand. If these three items were correctly identified, an extra 10 items were added (clothespin, comb, dice, screw, bolt, paperclip, rubber band, pen, pencil, coin). The Jamar^®^ Stereognosis Kit was used, including matching cards featuring a drawing of the item to point at the item recognized, as recommended [20,22]. The total number of correctly identified items was the final score (i.e., score 0–13).

Proprioception was measured using a thumb-wrist-position task (TWPT): The therapist passively moved the participant’s wrist into dorsal or palmar flexion and the MCP joint into extension or flexion with the participant’s vision blocked. Subjects were required to verbally indicate the end joint position. Outcomes were rated as 1 (unable), 2 (unable to identify either wrist or thumb end position), or 3 (able to correctly identify end position of both thumb and wrist).

In addition to the TWPT used, proprioception was also measured using a thumb localization task (TLT): With the participant’s vision blocked, the therapist passively moved the participant’s non-dominant upper limb laterally from the midline. The participant was asked to pinch the thumb of the non-dominant hand with the thumb and index finger of their dominant hand [23]. The task was scored as 1 (unable to locate thumb position), 2 (difficulty locating thumb position), or 3 (no difficulty locating thumb position).

Data were collected at three different time points. Baseline measurements were collected 14 days before the initial start of the therapy program (PRE). Post-intervention measurements were taken on the last day of the program (POST). Follow-up measurements were taken at six months follow-up (FOLLOW-UP). At all three measurement times, the child was assessed by the same therapist, not the child’s own therapist. Therapists were not blinded to the time of measurement.

### 2.4. Secondary Outcome Measures

Goal Attainment Scaling (GAS) is an evaluative tool to assess individual treatment and/or intervention goals achieved during/after an intervention. The GAS consists of a 6-point scale, ranging from −3 to +2. A score of −2 represents the participant’s performance at baseline, and improvements in the performance of the goal are scored ranging from −1 to +2, where the 0 score corresponds to the expected outcome and a score of −3 reflects deterioration [24]. GAS has shown to be a sensitive and valid method for defining motor function goals and shows excellent intra- and inter-rater reliability [25,26]. Changes of two points or more are defined as a clinically relevant difference [27]. For each participant, the most important rehabilitation goal was translated into a GAS by the participant’s assigned therapist from Adelante. Predetermined criteria for the progress towards that specific rehabilitation goal were defined. The individual’s performance of this goal was filmed and scored by therapists who have vast expertise in working with children with unilateral spastic CP and who are specialized in hand function problems in children. The video recordings of one participant were scored by the same therapist, not the child’s own therapist, on all three measurement time points. Therapists were not blinded to previous outcomes.

Canadian Occupational Performance Measure (COPM) is a semi-structured interview in which participants identify and rank their perceived hand function problems in everyday bimanual activities. The approach of this measure corresponds to the goal-oriented approach of the therapy program. The primary problem corresponds to the most important rehabilitation goal. Participants were asked to rate their performance and satisfaction for each problem on a 10-point scale, resulting in a mean total performance and satisfaction score. The COPM has good construct, content, and criterion validity. Test-retest reliability is high (0.76–0.89), and other ICC values of reliability remain to be tested in this population [28]. COPM performance and satisfaction were scored by the participants. Changes of two points or more were classified as a clinically relevant difference [29].

Assisting Hand Assessment (AHA) is an evaluative tool to rate bimanual performance. The AHA assesses the spontaneous use of the impaired hand in bimanual activities during a semi-structured activity session, which is video recorded. Afterwards, 22 items describing object-related hand actions were scored and converted to 0–100 logit-based AHA units [30]. We used the AHA 18–18 ‘Go-with-the-Floe’ board game in most cases, and in some participants, the Present task. A change of ±5 AHA units is considered the smallest detectable difference [30,31]. The video recordings of the participant’s performance on the AHA were scored by a selected group of therapists trained in the scoring of these videos. AHA scores of one participant were scored by the same therapist, not the child’s own therapist, on all three measurement time points. Therapists were not blinded to previous outcomes.

### 2.5. Data Analyses

Non-parametric statistics were used. To test overall improvement, a one-way analysis of variance by ranks (Friedman) was applied with post hoc Wilcoxon analyses when a significant effect was found. Data were analyzed using IBM SPSS Statistics version 27 (IBM SPSS Statistics, IBM Inc., New York, NY, USA). Statistical significance was set at α = 0.05. Multiple comparisons included the Bonferroni correction to avoid spurious false positives.

## 3. Results

### 3.1. Participants

Twenty-four children, 14 boys (58.3%) and 10 girls (41.7%), were analyzed. In 2017, seven participants were included, five in 2018, five in 2019, three in 2020, and four participants in 2021. The mean age at baseline was 14.21 years (±1.62), ranging from 12 to 17 years. Based on MRI data, we were able to perform a classification of 22 of the participants. Patient characteristics of the study sample are presented in Table 1.

### 3.2. Error Analysis

Post-intervention measurements were taken on the last day of the program so participants did not have to return to the rehabilitation center, especially for this. In some cases, however, this led to time constraints for the therapists, which meant they had to make choices about which test could be taken and which could not. This led to missing values (time). A COVID infection in two children at follow-up also caused missing data (COVID). One participant was hospitalized at follow-up (hospital). The reason for hospitalization was not related to arm-hand function impairments or participation in our program. An overview of the number of missing values per test on each of the three measurement points is given in Table 2.

### 3.3. Somatosensory Function

Changes in clinical somatosensory outcomes are summarized in Table 3.

The participants showed increased scores on the somatosensory function tests over time (*p* < 0.001), except for the thumb localization task. Multiple comparisons showed that participants scored significantly better on the stereognosis task, monofilament task, and vibration task between the PRE and POST measurements and between the PRE and FOLLOW-UP measurements. Differences between POST and FOLLOW-UP scores were not statistically significant.

### 3.4. GAS

GAS scores on the most important treatment goal, post-intervention and at follow-up, are shown in Figure 1.

Overall, participants showed a better GAS score on the primary treatment goal over time (*p* < 0.001). All participants, except one, exceeded the minimal clinical important difference (MCID) of 2 points. Post hoc pairwise comparisons revealed a significant difference between baseline and post-intervention (*p* < 0.001) and between baseline and follow-up (*p* < 0.001). The difference in GAS scores between post-intervention and follow-up was not significant (*p* = 0.480).

A list of the individual treatment goals for all participants is provided as Appendix A.

### 3.5. COPM

In total, 113 rehabilitation goals (17 × 5 + 7 × 4) were formulated using the COPM. Seventeen participants formulated five goals, whereas seven participants formulated four goals. The median and inter-quartile ranges of the weighted average COPM performance and satisfaction scores, for all time points, are depicted in Figure 2 and Figure 3. It should be noted here that not all children were able to score their satisfaction with the achievement of their treatment goal at baseline (*n* = 6).

Participants gave better COPM performance scores over time (*p* < 0.001). Post hoc pairwise comparisons revealed a significant difference between pre- and post-intervention (*p* < 0.001) and between pre-intervention and follow-up (*p* < 0.001). The difference between post-intervention and follow-up was not significant (*p* = 0.064).

COPM satisfaction scores were significantly better over time (*p* < 0.001). Post hoc analysis showed a significant increase in COPM satisfaction between PRE and POST scores, as well as between PRE and FOLLOW-UP scores (*p* < 0.001). COPM satisfaction scores between POST and FOLLOW-UP reveal a significant decrease (*p* = 0.007).

### 3.6. AHA

The unit scores on the AHA over time are shown in Figure 4.

Participants showed better AHA unit scores over time (*p* < 0.001). Post-intervention, 17/24 participants improved clinically meaningfully on the AHA with five or over five units, and 7/24 improved, but under five units. Multiple comparisons showed an increase in AHA scores between pre- and post-intervention (*p* < 0.001) and between pre-intervention and follow-up (*p* < 0.001). The difference in AHA scores between post-intervention and follow-up was not significant (*p* = 0.050).

## 4. Discussion

Although the somatosensory function is important for motor output, changes in somatosensory function associated with rehabilitation interventions have been understudied. The current study aimed to assess the effectiveness of a 15-day intensive functional clinical therapy program on somatosensory function. This program was focused on improving individual bimanual goals in children and adolescents with CP, GMFCS classification I-IV, MACS I-III, aged between 11–20 years, unilateral or asymmetric bilateral CP, and clear treatment goals on bimanual performance tasks. A significant improvement in personal goals and a significant improvement in the AHA were found, suggesting improved hand use of the more affected hand during bimanual task performance. Our results are consistent with the findings in the literature, indicating that intensive bimanual therapy leads to improvements in the bimanual performance of children and adolescents with USCP [32]. However, more importantly, somatosensory hand function also appears to improve as a result of this type of intervention. Even though no tactile-directed training nor exposure to special tactile-enriched materials were given during the intervention period, significant improvements were observed on all but one somatosensory test.

### 4.1. Improvements in Primary and Secondary Outcome Measures

Vibration sense, tactile perception (measured by a modified version of the monofilament test), stereognosis, and proprioception (measured by the thumb-wrist position task) improved during the program. This effect was retained at six months follow-up. Regarding proprioception, only a main effect of the intervention was found. Proprioception, as measured with the thumb localization task, showed no significant improvement. It is worthwhile noticing that at baseline, 74% of the participants already performed at the maximum level on this task. In hindsight, the thumb localization test may be less appropriate to investigate possible impairments in proprioception in children with CP. These clinical tools to quantify proprioception are known to lack sensitivity to small changes, offer poor reliability, and carry the potential for examiner bias [33].

The improvements in tactile perception and stereognosis, however, are promising findings, as they are essential for the dexterous manipulation of objects [7,34] and activities of daily life [35]. Somatosensory contributions to motor control in children with CP have been investigated [36,37,38], and it is acknowledged that somatosensory impairment has an effect on motor impairments in children with CP [7]. As mentioned before, studies on the effects of rehabilitation interventions on somatosensory hand function in children with CP are scarce and reveal varying results. In terms of intervention dose and content, our clinical program is most similar to the HABIT program, studied by Kuo and colleagues [13]. In their study, twenty children with USCP were randomized to receive either bimanual therapy (HABIT) or HABIT plus tactile training using tactile stimulating materials without vision. The HABIT group received the same dosage of training with the same material, but without specific tactile-directed training, i.e., standardized HABIT-full vision. Both groups improved on the grating orientation task, while stereognosis of the more-affected hand tended to improve, and no changes were found in the two-point discrimination task and monofilaments. Saussez et al. [14] investigated a similar intervention also involving the lower-extremities (HABIT-ILE) and without sensory enriched environment. They showed an improvement in stereognosis in the more affected hand in CP subjects, but no significant change in tactile spatial discrimination. Even though no additional sensory-enriched material or specific tactile training was used in our program, changes in tactile perception were found in addition to the improvement in stereognosis. These findings may be explained by the fact that participants were trained extensively in sensorimotor integration rather than just in sensory abilities. After all, goal-directed movements of the hand, which are necessary to perform most tasks of daily living, involve interacting with and manipulating objects in the environment and rely on sensorimotor integration. Intensive hand therapy is known to induce neuroplastic changes. Functional Magnetic Resonance Imaging (fMRI) or magnetoencephalography (MEG) after CIMT [39] or HABIT [40] demonstrated increased activation in the primary somatosensory cortex and an increase in the activation and size of the motor areas controlling the affected hand. Jobst et al. [11] were the first to demonstrate simultaneous improvement in sensory tactile registration in the affected hand and enhanced sensory processing in the contralateral primary somatosensory cortex after CIMT in children with hemiplegic CP.

In addition to improvement in somatosensory hand function, intensive bimanual training resulted in improvements in performance of personal treatment goals, as reflected by a higher GAS score, indicating a better execution of these activities in daily life. Most participants reached or exceeded their expected performance level, and the effect was retained at follow-up. Similar effects were observed regarding personal rehabilitation goals, as gauged using the COPM [41,42]. This finding is promising, since the improvement in the execution of activities in daily life settings results in an increase in independence, benefitting participation and quality of life. We assume that training and focusing on improving child-specific functional goals enhances the performance of these tasks in daily life. The satisfaction scores of the COPM increased significantly immediately after BIMT, though a slight decrease was observed at six months follow-up. A recent report by Figueiredo et al. [43] showed similar results in a group of children with bilateral CP participating in an intensive 90 h bimanual training program called HABIT. The group that performed the program exhibited greater improvements in performance and satisfaction with the performance of functional goals and functional skills than children who maintained their customary care routines and are following the results found by Bleyenheuft et al. [44]. It is acknowledged that targeting daily activities in individuals with CP is important [45,46]. Caregivers’ priorities of children with CP report on activities of daily living, especially self-care, to be the most frequent functional priority [46,47], while children’s ability to perform self-care activities facilitates socialization with peers, participation in community activities, and transition to independent living, as well as a reduction in caregiver’s burden [46]. Intensive goal-oriented bimanual training of participants, with the engagement of their caregivers, may play an important role in addressing this outcome.

Spontaneous use of the affected hand in bimanual performance tasks (AHA) improved to a clinically important difference after the program, and this effect was retained at follow-up. These findings are in line with the findings of the above-mentioned study by Kuo et al. [13].

### 4.2. Limitations

Direct comparison of our study results with the literature is difficult because of the large variability in study design, patient characteristics, and sensory assessment methods used. Even when the same sensory test is used, the method of administration and/or the method of scoring may differ. To improve our understanding of somatosensory hand function, we advocate for an international consensus on a clinically relevant core set with uniformity in methodology and scoring.

In our study group, 20.8% of participants have cortical malformation, which is known to result in less severe hand impairments than white or grey matter damage [48]. White matter lesions, in turn, lead to less severe hand function problems than grey matter lesions [15]. Participants with white matter damage and grey matter damage were approximately equally distributed in our study group. However, the relatively small number of participants does not allow for an extensive multivariate analysis to look for group differences.

Our study is a non-blinded observational cohort study, lacking a formal control group. Therefore, the level of evidence of the efficacy of our program is limited (level III on the Oxford levels of evidence). However, studies reporting on the efficacy of intensive bimanual training on sensory function in children and adolescents with CP are limited. Therefore, this study adds to the evidence that functional, intensive, goal-directed therapy may be effective in improving somatosensory hand function. Future studies including a larger number of participants and a control condition (e.g., waiting list control group) are needed to show the efficacy at a higher level of evidence, though complete blinded randomization can never be accomplished due to the nature of the intervention.

Finally, we used a clinical measurement protocol to measure somatosensory hand function. This protocol takes into account the testing time and burden for the participants, because motor outcome measures at the ICF activity level were recorded in addition to somatosensory measures. The advantage of this protocol is that it allows for proper assessment of the participant’s hand function, and a disadvantage is that it deviates slightly from the original protocol, especially regarding the Semmes-Weinstein monofilament test. The fact that we asked the participants to point to the location of touch may have influenced the reliability of the test [18].

## 5. Conclusions

Somatosensory function, including tactile perception, vibration sense, stereognosis, and position sense of the more affected hand in children and adolescents with unilateral spastic CP may improve after bimanual intensive functional training, without environmental tactile enrichment. A possible explanation for this might be that goal-directed training enhances sensorimotor integration.

## Figures and Tables

**Figure 1 jcm-12-01595-f001:**
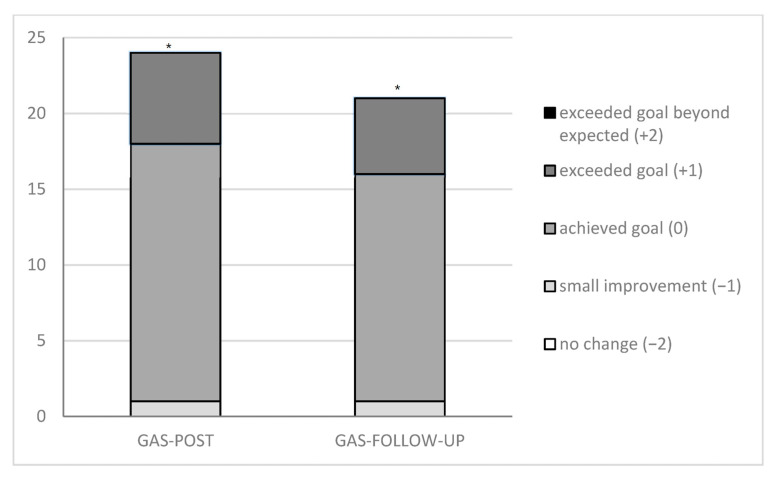
GAS score POST intervention and at FOLLOW-UP. Abbreviations: GAS = Goal Attainment Scaling; POST = directly after the program; FOLLOW-UP = six months after the program; * = significant difference *p* < 0.016 (Wilcoxon signed-rank test).

**Figure 2 jcm-12-01595-f002:**
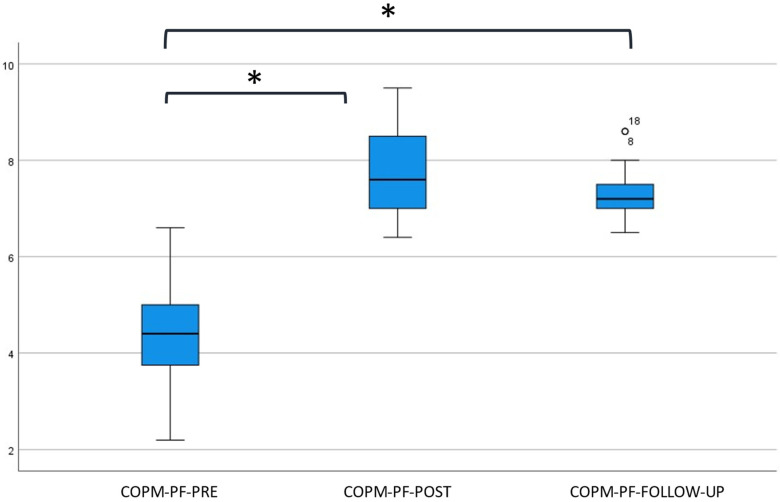
Boxplot of the weighted mean COPM performance score at baseline, post-intervention, and follow-up. Abbreviations used: COPM = Canadian Occupational Performance Measure; PF = Performance; PRE = baseline measurements; POST = directly after the program; FOLLOW-UP = six months after the program; * = significant difference *p* < 0.016 compared to baseline (Wilcoxon signed-rank test); circle identifies ouliers.

**Figure 3 jcm-12-01595-f003:**
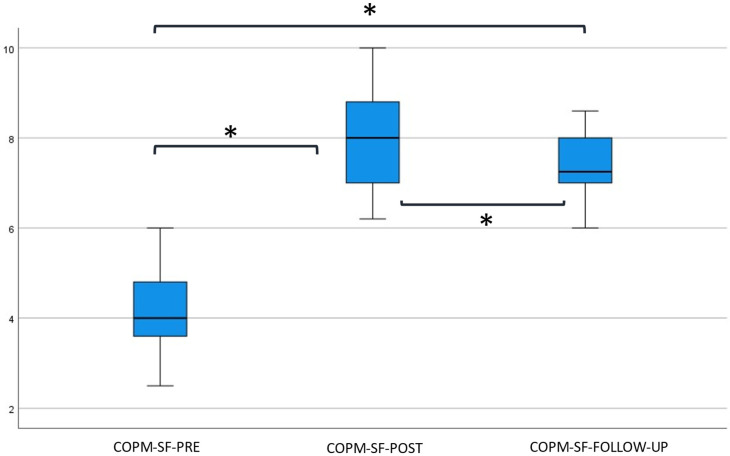
Boxplot of the weighted mean COPM satisfaction score at baseline, post-intervention, and follow-up. Abbreviations used: COPM = Canadian Occupational Performance Measure; SF = Satisfaction; PRE = baseline measurements; POST = directly after the program; FOLLOW-UP = six months after the program; * = significant difference *p* < 0.016 compared to baseline (Wilcoxon signed-rank test).

**Figure 4 jcm-12-01595-f004:**
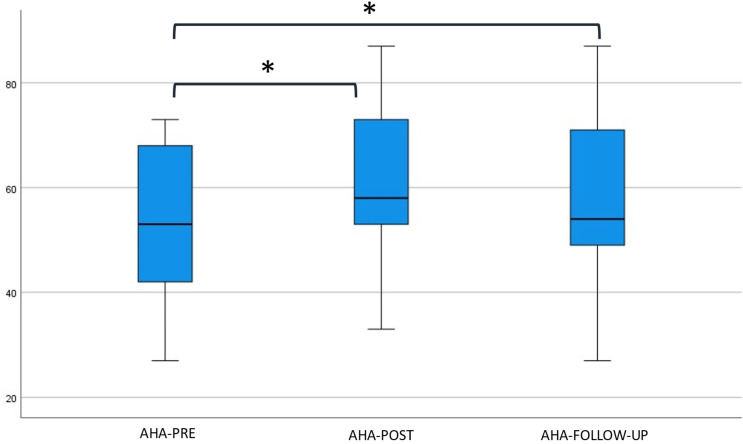
AHA unit scores at baseline, post-intervention, and follow-up. Abbreviations used: AHA = Assisting Hand Assessment; PRE = baseline measurements; POST = directly after the program; FOLLOW-UP = six months after the program; * = significant difference *p* < 0.016 (Wilcoxon signed-rank test).

**Table 1 jcm-12-01595-t001:** Descriptive data of participants (*n* = 24).

Characteristics	Group
Age (years) (mean, SD)	14.21 (±1.62)
Gender (*n* (%))	
Male	14 (58.3)
Female	10 (41.7)
Type of CP (*n* (%))	
Unilateral spastic	23 (95.8)
Bilateral spastic	1 (4.2)
MACS (*n* (%))	
I	9 (37.5)
II	12 (50)
III	3 (12.5)
GMFCS (*n* (%))	
I	20 (83.3)
II	4 (16.7)
Paretic hand (*n* (%))	
Right	15 (62.5)
Left	9 (37.5)
Lesion type (*n* (%))	
WMI	8 (33.3)
CM	5 (20.8)
GMI	9 (37.5)
Unknown	2 (8.3)

Abbreviations used: MACS = Manual Ability Classification System; GMFCS = Gross Motor Function Classification System; GMI = Gray matter injury; WMI = white matter injury; CM = cortical malformation.

**Table 2 jcm-12-01595-t002:** The number of missing values per test, per measurement point.

	PRE *n* Missings	POST *n* Missings	FOLLOW-UP *n* Missings
TLT	1→1 (time)	1→1 (time)	4→1 (time) 2 (COVID) 1 (hospital)
SGT	0	0	3→2 (COVID) 1 (hospital)
MFT	0	0	3→2 (COVID) 1 (hospital)
TWPT	1→1 (time)	1→1 (time)	4→1 (time) 2 (COVID) 1 (hospital)
VT	1→1 (time)	0	3→2 (COVID) 1 (hospital)
GAS	0	0	3→2 (COVID) 1 (hospital)
COPM	0	0	3→2 (COVID) 1 (hospital)
AHA	0	0	3→2 (COVID) 1 (hospital)

Abbreviations used: TLT = thumb localization task; SGT = stereognosis task; MFT = monofilament task; TWPT = thumb wrist position task; VT = vibration task; GAS = Goal Attainment Scaling; COPM = Canadian Occupational Performance Measure; AHA = Assisted Hand Assessment.

**Table 3 jcm-12-01595-t003:** Number and percentage of participants and their test scores at baseline, directly after BIMT, and six months follow-up.

	Score	PRE *n* (%)	POST *n* (%)	FU *n* (%)	Overall Difference *p*	PRE-POST Bonferroni Difference *p*	POST-FU Bonferroni Difference *p*	PRE-FU Bonferroni Difference *p*
TLT	1	3 (13)	3 (13)	2 (10)	0.076	0.317	0.180	0.059
	2	3 (13)	1 (4.3)	0 (0)				
	3	17 (74)	19 (82.7)	18 (90)				
SGT	0	13 (54.2)	7 (29.2)	7 (33.3)	0.018 *	<0.001 *	0.411	0.009 *
	1–3	7 (29.1)	9 (37.4)	5 (23.8)				
	≥4	4 (16.7)	8 (33.3)	9 (42.9)				
MFT	1	7 (29.2)	16 (66.7)	14 (66.7)	<0.001 *	<0.001 *	0.317	<0.001 *
	2	11 (45.8)	8 (33.3)	7 (33.3)				
	3	6 (25)	0 (0)	0 (0)				
TWPT	1	3 (13)	1 (4.3)	1 (5)	<0.001 *	0.059	1.00	0.059
	2	2 (8.7)	1 (4.3)	1 (5)				
	3	18 (78.3)	21 (91.3)	18 (90)				
VT	0	3 (13)	0 (0)	0 (0)	<0.001 *	0.003 *	0.180	0.007 *
	1–10	5 (21.7)	0 (0)	1 (4.7)				
	11–18	15 (65.3)	24 (100)	20 (95.3)				

Abbreviations used: TLT = thumb localization task; SGT = stereognosis task; MFT = monofilament task; TWPT = Thumb wrist position task; VT = vibration task; PRE = baseline; POST = directly after the program; FU = Follow-up six months after the program; * significant difference.

## Data Availability

The data are not publicly available due to privacy restrictions. The data presented in this study are available on request from the corresponding author.

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
