# Peer review of "The Effect of Bimanual Intensive Functional Training on Somatosensory Hand Function in Children with Unilateral Spastic Cerebral Palsy: An Observational Study"

_jcm, 2023, doi:10.3390/jcm12041595_

Round 1

Reviewer 1 Report (Previous Reviewer 2)

I am happy with this manuscript as it is.

Congratulations!

Author Response

Thank you for your time and effort to review our manuscript, and thank you for your compliments.

Reviewer 2 Report (New Reviewer)

The authors investigated the effectiveness of a 15-days functional, intensive, goal-oriented, clinical therapy program, focused on improving bimanual performance in daily activities in children and adolescents with CP, aged between 12-17 years, on somatosensory performance.

The results indicate that BIMT may improve somatosensory function of the more affected hand in children with unilateral spastic CP. It is an examination of a clinical question that has not yet been fully elucidated, which is interesting.

On the other hand, the number of subjects was small and the background of the subjects were heterogeneous, so the results should be interpreted with caution.

Major points

#1 I wonder if intensive training itself in the extreme chronic phase of CP can improve long-term somatosensory function. Had he or she never had the motivation to use both hands in daily life by the age? It is hard to imagine that a child with CP has not received any physical or occupational therapy until adolescent. Since these investigations are relative results compared to pre-BIMT, post-BIMT and follow up period, it is very important to inform that what therapy and the intensity the subjects received before BIMT.

The authors should add information on the content and intensity of the therapy received before BIMT, and add the discussion based on that as well.

#2 The numbers of n in Tables 2 and 3 do not match.

total n=24

TLT: PRE missing n=0 in Tble2 but PRE n=23 in Table3.

MFT: POST missing n=1 in Tble2 but POST n=24 in Table3

TWPT: POST missing n=1 in Tble2 but POST n=24 in Table3

VT: PRE missing n=0 in Tble2 but PRE n=23 in Table3

These tables are the main results of this study and these discrepancies undermine the credibility of the analysis.

Minor point

#3 Figure 1 is a figure that is difficult to distinguish visually. Color figure can be used, so it is better to use color figures.

Author Response

The authors investigated the effectiveness of a 15-days functional, intensive, goal-oriented, clinical therapy program, focused on improving bimanual performance in daily activities in children and adolescents with CP, aged between 12-17 years, on somatosensory performance.

The results indicate that BIMT may improve somatosensory function of the more affected hand in children with unilateral spastic CP. It is an examination of a clinical question that has not yet been fully elucidated, which is interesting.

On the other hand, the number of subjects was small and the background of the subjects were heterogeneous, so the results should be interpreted with caution.

We totally agree with the reviewer, our results are promising, however, as we pointed out under ‘study limitations’, further research including a larger number of participants and a control condition (e.g. waiting list control group) is needed to show the efficacy at a higher level of evidence.

Major points

#1 I wonder if intensive training itself in the extreme chronic phase of CP can improve long-term somatosensory function. Had he or she never had the motivation to use both hands in daily life by the age? It is hard to imagine that a child with CP has not received any physical or occupational therapy until adolescent. Since these investigations are relative results compared to pre-BIMT, post-BIMT and follow up period, it is very important to inform that what therapy and the intensity the subjects received before BIMT.

The authors should add information on the content and intensity of the therapy received before BIMT, and add the discussion based on that as well.

As indicated in the ‘Patient’ section on page 4, all participants included in this study, participated in an intensive bimanual therapy programme for the first time. However, this does not mean they did not receive therapy until adolescent. All participants received therapy as usual. In the Netherlands, usual care consists of 1 or 2 times a week 30 min, based on request for help from the parents and/or children themselves or in response to the findings of their annual screening. This is in line with the Dutch guideline for the treatment of children with CP. This information was added to the text in section ‘2.2 Bimanual intensive functional training’ on page 5.

#2 The numbers of n in Tables 2 and 3 do not match.

total n=24

TLT: PRE missing n=0 in Tble2 but PRE n=23 in Table3.

MFT: POST missing n=1 in Tble2 but POST n=24 in Table3

TWPT: POST missing n=1 in Tble2 but POST n=24 in Table3

VT: PRE missing n=0 in Tble2 but PRE n=23 in Table3

These tables are the main results of this study and these discrepancies undermine the credibility of the analysis.

We thank the reviewer for this perceptiveness. To avoid any misunderstanding, we have meticulously re-checked the entire database for missing values, and we can guarantee that the numbers as they appeared in Table 3 were correct. The numbers in Table 2 are adjusted according to the correct numbers in Table 3.

TLT: PRE missing n=1, adjusted in Table 2

MFT: POST missing n=0, adjusted in Table 2

TWPT: POST n=23 in Table 3, POST missing n=1 in Table 2, no correction made

VT: PRE missing n=1, adjusted in Table 2

However, when re-checking the data we discovered an inconsistency in subject #21. This subject had no measurements at follow-up, sensibility data were therefore correctly classified as missing, but the secondary outcome measures, i.e. the GAS, COPM and AHA, were not missing. We therefore removed these data from the database and redone all analyses in terms of secondary outcome measures. This only slightly affected p values, but it did not alter any statistical significance.

For this reason figures 1 to 4 were adjusted, as well as the p values in the text where applicable.

Minor point

#3 Figure 1 is a figure that is difficult to distinguish visually. Color figure can be used, so it is better to use color figures.

Figure 1 has been modified to make it easier to interpret visually.

Round 2

Reviewer 2 Report (New Reviewer)

This manuscript has been revised appropriately.

Author Response

We would like to thank the reviewer very much for his/her time and effort in reviewing our modified article. 

Comment:

This manuscript has been revised appropriately.

Thank you for reviewing our manuscript and approval. 

This manuscript is a resubmission of an earlier submission. The following is a list of the peer review reports and author responses from that submission.

Round 1

Reviewer 1 Report

Review of: The effect of bimanual intensive functional training on sensory hand function in children with unilateral spastic cerebral palsy.

Strengths
The objective of this study was to investigate whether bimanual intensive functional therapy without using enriched sensory materials improves somatosensory hand function. The strength of this manuscript is that it addresses a topic that is of interest to the medical community. The manuscript could be strengthened by: (1) an updated literature search on the effect of intensive training on somatosensory impairments of children with cerebral palsy to inform the introduction and discussion sections, (2) more details on the intervention, (3) more details on the psychometrics and scoring of the outcome measures.

Specific Recommendations:

Page 1, line 40 and page 10, line 317. The reference Auld et al., 2014 is not in the correct format and is not in the reference list.

Page 1, line 42. The introduction states that only two research groups reported on somatosensory impairments following intensive training, however, a quick PubMed search resulted in three others: Saussez et al, Journal of Child Neurology, 2018; Matusz et al, Neural Plasticity, 2018; Maitre et al, Brain Topography, 2020. An updated literature search may be indicated to determine all the research literature on the effect of intensive training on somatosensory impairments of children with cerebral palsy to inform the introduction and discussion sections.

Page 2, lines 62-64. Please include the number of participants in each BIMT program since the intervention occurred over 5 years.

Page 2-3, lines 92-99. Please include details of the content and dose of intervention using the TIDieR guidelines in the manuscript or in supplementary information. This is important not only for replication, but also because in the discussion (Page 10, lines 328-331) an explanation is given as to the reason the children in this study improved tactile perception whereas they did not improve in the Ko et al study. More details on the intervention are needed to support this statement.

Page 3, lines 100-131. Please provide information on the psychometric properties of each sensory function test if available.

Page 3, line 143-144. Please provide interrater reliability data on the scoring of the video data for the Goal Attaining Scaling measure. In supplementary information, please provide the list of intervention goals for all participants.

Page 4, lines 155-161. Please state who scored the videos of the Assisting Hand Assessment.

Page 3-4, lines 100-161. Please state whether the post-intervention assessors were masked to the pre-intervention scores for all outcome assessments.

Page 9, Discussion. If more trials are found in the updated literature search on the effect of intensive training on somatosensory impairments of children with cerebral palsy, please add them to the discussion section.

Please add the clinical trial protocol or clinical trial registration, if available.

Reviewer 2 Report

I find this an interesting and well written manuscript, focussing on sensory impairment and the improvement of this through functional training in a structured program. Congratulations!

The introduction is relevant.

Methods are sound and comprehensive, albeit somewhat modified regarding tactile testing. Activity level measures are also included.

Results are quite remarcable, and seem to be retained over time. 

A larger group with similar neuroimaging findings would be interesting to explore in the future. 

Reviewer 3 Report

This is a very poor study. I have many concerns about the method of this paper This manuscript has very poor language quality such that it cannot be understood by readers. The manuscript needs extensive revision for language and grammar

On the other hand the reliability of treatment is controversial. The details of the procedure are not mentioned. Students are responsible from the treatment and this is inappropriate and is a bias.  The results of the evaluations mentioned in the method were not given, many of the evaluations given as a result were not suitable for the purpose of the study and were not mentioned in the method.  I do not think that the results of this study are at a level that can contribute to the literature.